# Design of Longitudinal–Torsional Transducer and Directivity Analysis during Ultrasonic Vibration-Assisted Milling of Honeycomb Aramid Material

**DOI:** 10.3390/mi13122154

**Published:** 2022-12-06

**Authors:** Mingxing Zhang, Zuotian Ma, Xiaodong Wang, Ting Meng, Xiangqun Li

**Affiliations:** 1Engineering Training Center, Changchun University of Technology, Changchun 130012, China; 2School of Mechanical and Electrical Engineering, Changchun University of Technology, Changchun 130012, China; 3Changchun Yidong Clutch Co., Ltd., Changchun 130012, China

**Keywords:** honeycomb aramid material, longitudinal–torsional transducer, directivity, machining stability

## Abstract

This paper presents a longitudinal–torsional transducer for use during the ultrasonic vibration-assisted milling (UVAM) of honeycomb aramid material. The mechanism of longitudinal–torsional conversion was analyzed to guide the design of a vibration transducer. The transducer features five spiral grooves around the front cover plate, which function under the excitation of a group of longitudinal piezoelectric ceramics. A portion of the longitudinal vibration was successfully converted into torsional vibration. The resonant frequency, longitudinal vibration displacement and torsional amplitude at the top of the disk milling cutter were 24,609 Hz, 19 μm and 9 μm, respectively. In addition, the directivity of the longitudinal–torsional transducer was theoretically analyzed. Compared with conventional milling, UVAM with the longitudinal-torsional could significantly reduce the cutting force (40–50%) and improve the machining stability.

## 1. Introduction

Honeycomb aramid materials have been widely used in the field of aerospace because of their excellent structural and mechanical properties. However, honeycomb core deformation, the risk of hole grid collapse, and a low yield are among challenges faced by traditional machining methods in the machining of honeycomb aramid materials. Based on the resonance effect, ultrasonic vibration-assisted milling (UVAM) can generate a high-frequency relative motion between the tool and the workpiece, which is conducive to improving processing stability and surface quality.

At present, most ultrasonic transducers used in UVAM vibrate along the longitudinal direction. However, longitudinal vibration has a limited effect on the improvement in terms of workpiece machinability [1,2]. In order to improve the machining stability of honeycomb aramid materials during UVAM, this paper presents a longitudinal–torsional transducer, which can excite the milling cutter to vibrate in both longitudinal and torsional directions simultaneously. The longitudinal and torsional vibrations of the longitudinal–torsional transducer can be realized in two ways: the vibration mode conversion of the transducer and the tangential polarization of a piezoelectric ceramic stack.

In the process of vibration mode conversion, torsional vibrations in the transmission rod can be obtained based on the coupling and degradation of horn form longitudinal vibration. Patrick Harkness [3] proposed two design methods for longitudinal–torsional mode transitions. Celaya [4] proposed a booster that can improve the surface quality but with a low amplitude. Liu [5] demonstrated two forms of thread grooves which can degrade longitudinal vibrations into longitudinal–torsional vibrations. Yahya [6], Gao [7], Yuan [8], Zhang [9] and Wu [10] developed L–T transducers based on theoretical and simulation analysis and applied them to the processing of Ti-6Al-4V. Budairi [11] used numerical, analytical and experimental methods to evaluate transducer performance under different excitations based on the L–T response. In addition, by setting different degradation modes on the cylinder surface, the same method can be applied to ultrasonic motors [12,13,14]. Shen [15] conducted an assisted ultrasonic vibration micro-milling process on aluminum alloy material and used a combination of machining parameters to demonstrate that ultrasonic vibration results in a significant improvement in terms of surface roughness; Chen [16] came to the same conclusion for silicon carbide ceramic grinding. Cleary [17] and Wang [18] proposed an L–T ultrasound needle device for medical treatment and implantation applications which provides a potential application scenario for L–T mode devices. Numanoğlu [19] and Akbaş [20] determined the vibrations of composite rods and beams for different boundary conditions using different theoretical methods, providing a theoretical foundation for the derivation of an L–T transducer. However, in the processing of honeycomb aramid materials by UVAM, there is a lack of research on longitudinal–torsional transducers, which limits the possible improvements to be made regarding the machinability of honeycomb aramid materials.

The tangential polarization of a piezoelectric ceramic stack can be realized by combining axial piezoelectric ceramics and tangential piezoelectric ceramics. Kim [21] proposed a transducer with two frequencies by combining two piezoelectric torsional discs [22] designed a torsional device. Harada [23] used a torsional transducer and a microporous plate to generate droplets. However, due to the complex fabrication process associated with tangentially polarized piezoelectric wafers, problems such as electrical breakdown and incomplete polarization will occur when the chip size is large. Therefore, it is difficult to develop torsional vibration transducers with higher frequencies using the current level of ceramic production technology.

In addition, the stability lobes diagram is of great significance for the optimization of equipment structure and process parameters, with it being affected by the directivity of a transducer during UVAM. At present, the relevant research primarily focuses on analysis methods for stability [24], the correctness of the stability lobes diagram [25] and increasing the equivalent stiffness by filling the honeycomb material with viscoelastic damping material [26]. Suárez [27] showed that the application of ultrasonic milling can improve the form of tool wear patter and render the evolution more stable. However, the research on the directivity of transducers during ultrasonic vibration-assisted milling (UVAM) of honeycomb aramid material is relatively sparse.

This paper presents a longitudinal–torsional transducer for use during the ultrasonic vibration-assisted milling (UVAM) of honeycomb aramid material. To avoid energy loss, the front cover plate and the horn of transducer were designed as a whole, and spiral grooves were machined on the front cover plate. Based on the theory of longitudinal–torsional transformation, the spiral groove parameters on the front cover plate were analyzed, and the directivity of the longitudinal–torsional transducer was studied theoretically. Finally, the performance of the longitudinal–torsional transducer and its processing stability were verified by experiments.

## 2. Design of Longitudinal–Torsional Transducer

### 2.1. Theoretical Analysis of Longitudinal–Torsional Transducer

Ultrasonic transducers can be divided into two types according to different internal working mediums: sandwich piezoelectric ceramic transducers and magnetostrictive transducers [28]. Sandwich piezoelectric ceramic transducers are widely used because of the advantages of uniform vibration, a simple structure andease in terms of forming and processing, etc. This type of transducer, which is composed of two quarter wavelengths in length, as shown in Figure 1, was selected for this study. Where Z_i_ is the characteristic acoustic impedance of each part of the material, X_i_ is the boundary of each component, L_i_ is the length of each component (i = 1,2, …, 8).

According to the theory related to Newtonian dynamics in the principle of vibration transmission, the equation for vibration at any section can be expressed as:(1)$$\frac{\partial \left(S\cdot \sigma \right)}{\partial x}dx=S\cdot \rho \frac{\partial^{2}\xi }{\partial t^{2}}dx$$
where *S* is area function of the rod at any cross section, *S* = *S*(*x*), ξ is the displacement function of the mass, *σ* is the stress function of the rod, which can be represented as $\sigma =\sigma \left(x\right)=E\frac{\partial \xi }{\partial X}$ and *ρ* and *E* are the density and Young’s modulus of horn, respectively.

In the process of UVAM, the vibration form of the transducer is simple harmonic vibration. $\xi =\xi \cdot e^{j\omega t}$, $\kappa =\frac{\omega }{c}$ and *c* represent the circular frequency and propagation of the speed of sound wave, respectively. Therefore, the vibration equation for the transducer at any section can be written as follows:(2)$$\frac{\partial^{2}\xi }{\partial x^{2}}+\frac{1}{S}\cdot \frac{\partial S}{\partial x}\cdot \partial \frac{\partial \xi }{\partial x}+k^{2}\xi =0$$
where *k* represents the number of circular waves and *k* = *ω*/*c*. For metallic materials, $c=\sqrt{E/\rho }$; for piezoelectric ceramic materials, the equivalent sound velocity of the longitudinal wave is approximately expressed as: $c\approx c\sqrt{1-k_{33}^{2}}=\sqrt{1/(\rho S_{33}^{E})}$. Substituting *v* = *j**ω**t* into Equation (2), the vibration velocity for longitudinal vibration can be expressed as follows:(3)$$\frac{\partial^{2}v}{\partial x^{2}}+\frac{1}{S}\frac{\partial S}{\partial x}\cdot \frac{\partial v}{\partial x}+k^{2}v=0$$

Since *S*(*x*) is constant, Equation (3) can be simplified as:(4)$$\frac{\partial^{2}v}{\partial x^{2}}+kv^{2}=0$$

The general solution for the vibration velocity, i.e., Equation (4), can be obtained as:(5)$$v_{\left(x\right)}=A\sin kx+B\cos kx$$

Substituting Equation (5) into the stress distribution expression results in:(6)$$F_{\left(x\right)}=\frac{ES}{j\omega }\frac{\partial v}{\partial x}=-j\rho cs(A\cos kx-B\sin kx)=-jZ(A\cos kx-B\sin kx)$$
where *j* represents an imaginary sign *Z* is the characteristic acoustic impedance and Z=ρcs.

During the theoretical analysis of ultrasonic transducers, the system is usually divided into two parts using the nodal plane. In order to simplify the calculation, the nodal plane of an ultrasonic transducer was designed on the joint surfaces of the piezoelectric crystal stack and front cover plate. One part is composed of the rear cover plate, electrode plate and piezoelectric ceramic plate, and the other part is composed of the front cover plate and amplitude transformer, as shown in Figure 1. *Vf* and *Vb* are the vibration velocities of the end face of the rear cover plate and the front face of the piezoelectric ceramic plate, respectively. When the ultrasonic waves propagate at the medium partition interface, the boundary conditions on the left side of the nodal plane can be introduced as follows:(7)$$\begin{array}{l} v_{1}\left(0\right)=v_{b}\\ v_{1}\left(l_{1}\right)=v_{2}\left(0\right)\\ v_{2}\left(l_{2}\right)=v_{3}\left(0\right)\\ v_{3}\left(l_{3}\right)=v_{4}\left(0\right)\\ v_{4}\left(l_{4}\right)=v_{5}\left(0\right)\\ v_{5}\left(l_{5}\right)=0\\ F_{1}\left(0\right)=0\\ F_{1}\left(l_{1}\right)=F_{2}\left(0\right)\\ F_{2}\left(l_{2}\right)=F_{3}\left(0\right)\\ F_{3}\left(l_{3}\right)=F_{4}\left(0\right)\\ F_{4}\left(l_{4}\right)=F_{5}\left(0\right) \end{array}$$

Substituting Equation (7) into Equations (5) and (6), the following results can be obtained:(8)$$\begin{array}{l} A_{1}=0\\ A_{2}=-\frac{Z_{1}}{Z_{2}}v_{b}\sin k_{1}l_{1}\\ A_{3}=-\frac{Z_{1}}{Z_{3}}v_{b}\sin k_{1}l_{1}\cos k_{2}l_{2}-\frac{Z_{2}}{Z_{3}}v_{b}\cos k_{1}l_{1}\sin k_{2}l_{2}\\ A_{4}=\frac{Z_{1}}{Z_{4}}v_{b}\sin k_{1}l_{1}\left(\frac{Z_{3}}{Z_{2}}\sin k_{2}l_{2}\sin k_{3}l_{3}-\cos k_{2}l_{2}\sin k_{3}l_{3}\right)-\\ \;\;\;\;\;\;\;\;\;\;\frac{1}{Z_{4}}v_{b}\cos k_{1}l_{1}\left(Z_{2}\sin k_{2}l_{2}\cos k_{3}l_{3}+Z_{3}\cos k_{2}l_{2}\sin k_{3}l_{3}\right)\\ A_{5}=\frac{\cos k_{5}l_{5}}{\sin k_{5}l_{5}}v_{b}[\left(\frac{1}{Z_{4}}\cos k_{3}l_{3}\sin k_{4}l_{4}+\frac{1}{Z_{3}}\sin k_{3}l_{3}\cos k_{4}l_{4}\right)\\ \;\;\;\;\;\;\;\;\;\;\left(Z_{1}\sin k_{1}l_{1}\cos k_{2}l_{2}+Z_{2}\cos k_{1}l_{1}\sin k_{2}l_{2}\right)-(\frac{Z_{3}}{Z_{4}}\sin k_{3}l_{3}\sin k_{4}l_{4}-\\ \;\;\;\;\;\;\;\;\;\;\cos k_{3}l_{3}\cos k_{4}l_{4})\left(\frac{Z_{1}}{Z_{2}}\sin k_{1}l_{1}\sin k_{2}l_{2}-\cos k_{1}l_{1}\cos k_{2}l_{2}\right)] \end{array}$$
(9)$$\begin{array}{l} B_{1}=v_{b}\\ B_{2}=v_{b}\cos k_{1}l_{1}\\ B_{3}=-\frac{Z_{1}}{Z_{2}}v_{b}\sin k_{1}l_{1}\sin k_{2}l_{2}+v_{b}\cos k_{1}l_{1}\cos k_{2}l_{2}\\ B_{4}=-Z_{1}v_{b}\sin k_{1}l_{1}\left(\frac{1}{Z_{3}}\cos k_{2}l_{2}\sin k_{3}l_{3}+\frac{1}{Z_{2}}\sin k_{2}l_{2}\cos k_{3}l_{3}\right)+\\ \;\;\;\;\;\;\;\;\;\;v_{b}\cos k_{1}l_{1}\left(\cos k_{2}l_{2}\cos k_{3}l_{3}-\frac{Z_{2}}{Z_{3}}\sin k_{2}l_{2}\sin k_{3}l_{3}\right)\\ B_{5}=v_{b}\left(\frac{Z_{3}}{Z_{4}}\sin k_{3}l_{3}\sin k_{4}l_{4}-\cos k_{3}l_{3}\cos k_{4}l_{4}\right)(\frac{Z_{1}}{Z_{2}}\sin k_{1}l_{1}\\ \;\;\;\;\;\;\;\;\;\;\sin k_{2}l_{2}-\cos k_{1}l_{1}\cos k_{2}l_{2})-v_{b}(\frac{1}{Z_{4}}\cos k_{3}l_{3}\sin k_{4}l_{4}+\\ \;\;\;\;\;\;\;\;\;\;\frac{1}{Z_{3}}\sin k_{3}l_{3}\cos k_{4}l_{4})(Z_{1}\sin k_{1}l_{1}\cos k_{2}l_{2}+Z_{2}\cos k_{1}l_{1}\sin k_{2}l_{2}) \end{array}$$
where $Z_{i}$ is the characteristic acoustic impedance of each part of the material on the left side of the nodal plane and $Z_{i}=\rho cs_{i}$.

The vibration velocity and stress distribution of each part can be calculated as follows: (10)$$\begin{array}{l} v_{1}\left(x_{1}\right)=v_{b}\cos k_{1}x_{1}\\ v_{2}\left(x_{2}\right)=-\frac{Z_{1}}{Z_{2}}v_{b}\sin k_{1}l_{1}\sin k_{2}x_{2}+v_{b}\cos k_{1}l_{1}\cos k_{2}x_{2}\\ v_{3}\left(x_{3}\right)=v_{b}\sin k_{3}x_{3}(-\frac{Z_{1}}{Z_{3}}\sin k_{1}l_{1}\cos k_{2}l_{2}-\frac{Z_{2}}{Z_{3}}\cos k_{1}l_{1}\sin k_{2}l_{2})+\\ \;\;\;\;\;\;\;\;\;\;\;\;\;\;\;\;\;v_{b}\cos k_{3}x_{3}(-\frac{Z_{1}}{Z_{2}}\sin k_{1}l_{1}\sin k_{2}l_{2}+\cos k_{1}l_{1}\cos k_{2}l_{2})\\ v_{4}\left(x_{4}\right)=\sin k_{4}x_{4}[\frac{Z_{1}}{Z_{4}}v_{b}\sin k_{1}l_{1}\left(\frac{Z_{3}}{Z_{2}}\sin k_{2}l_{2}\sin k_{3}l_{3}-\cos k_{2}l_{2}\cos k_{3}l_{3}\right)-\\ \;\;\;\;\;\;\;\;\;\;\;\;\;\;\;\;\;\frac{1}{Z_{4}}v_{b}\cos k_{1}l_{1}\left(Z_{2}\sin k_{2}l_{2}\cos k_{3}l_{3}+Z_{3}\cos k_{2}l_{2}\sin k_{3}l_{3}\right)]+\cos k_{4}x_{4}\\ \;\;\;\;\;\;\;\;\;\;\;\;\;\;\;\;\;[v_{b}\cos k_{1}l_{1}(\cos k_{2}l_{2}\cos k_{3}l_{3}-\frac{Z_{2}}{Z_{3}}\sin k_{2}l_{2}\sin k_{3}l_{3})-Z_{1}v_{b}\sin k_{1}l_{1}\\ \;\;\;\;\;\;\;\;\;\;\;\;\;\;\;\;\;\left(\frac{1}{Z_{3}}\cos k_{2}l_{2}\sin k_{3}l_{3}+\frac{1}{Z_{2}}\sin k_{2}l_{2}\cos k_{3}l_{3}\right)]\\ v_{5}\left(x_{5}\right)=v_{b}\left(-\frac{\cos k_{5}l_{5}}{\sin k_{5}l_{5}}\sin k_{5}x_{5}+\cos k_{5}x_{5}\right)[\frac{Z_{3}}{Z_{4}}\sin k_{3}l_{3}\sin k_{4}l_{4}-\\ \;\;\;\;\;\;\;\;\;\;\;\;\;\;\;\;\;\cos k_{3}l_{3}\cos k_{4}l_{4})(\frac{Z_{1}}{Z_{2}}\sin k_{1}l_{1}\sin k_{2}l_{2}-\cos k_{1}l_{1}\cos k_{2}l_{2})-(\frac{1}{Z_{2}}\cos k_{3}l_{3}\\ \;\;\;\;\;\;\;\;\;\;\;\;\;\;\;\;\;\sin k_{4}l_{4}+\frac{1}{Z_{3}}\sin k_{3}l_{3}\cos k_{4}l_{4})(Z_{1}\sin k_{1}l_{1}\cos k_{2}l_{2}+Z_{2}\cos k_{1}l_{1}\sin k_{2}l_{2})] \end{array}$$
(11)$$\begin{array}{l} F_{1}\left(x_{1}\right)=jZ_{1}v_{b}\sin k_{1}l_{1}\\ F_{2}\left(x_{2}\right)=jZ_{2}v_{b}\left(\frac{Z_{1}}{Z_{2}}\cos k_{2}x_{2}\sin k_{1}l_{1}+\sin k_{2}x_{2}\cos k_{1}l_{1}\right)\\ F_{3}\left(x_{3}\right)=jZ_{3}v_{b}[\cos k_{3}x_{3}(\frac{Z_{1}}{Z_{3}}\sin k_{1}l_{1}\cos k_{2}l_{2}+\frac{Z_{2}}{Z_{3}}\cos k_{1}l_{1}\sin k_{2}l_{2})+\\ \;\;\;\;\;\;\;\;\;\;\;\;\;\;\;\;\;\sin k_{3}x_{3}\left(-\frac{Z_{1}}{Z_{2}}\sin k_{1}l_{1}\sin k_{2}l_{2}+\cos k_{1}l_{1}\cos k_{2}l_{2}\right)]\\ F_{4}\left(x_{4}\right)=jZ_{4}v_{b}\cos k_{4}x_{4}[-\frac{Z_{1}}{Z_{4}}\sin k_{1}l_{1}(\frac{Z_{3}}{Z_{2}}\sin k_{2}l_{2}\sin k_{3}l_{3}-\cos k_{2}l_{2}\cos k_{3}l_{3})+\\ \;\;\;\;\;\;\;\;\;\;\;\;\;\;\;\;\;\frac{1}{Z_{2}}\cos k_{1}l_{1}\left(Z_{2}\sin k_{2}l_{2}\cos k_{3}l_{3}+Z_{3}\cos k_{2}l_{2}\sin k_{3}l_{3}\right)]-jZ_{4}v_{b}\sin k_{4}x_{4}\\ \;\;\;\;\;\;\;\;\;\;\;\;\;\;\;\;\;[Z_{1}\sin k_{1}l_{1}\left(\frac{1}{Z_{3}}\cos k_{2}l_{2}+\frac{1}{Z_{2}}\sin k_{2}l_{2}\cos k_{3}l_{3}\right)-\cos k_{1}l_{1}(\cos k_{2}l_{2}\cos k_{3}l_{3}-\\ \;\;\;\;\;\;\;\;\;\;\;\;\;\;\;\;\;\frac{Z_{2}}{Z_{3}}\sin k_{2}l_{2}\sin k_{3}l_{3})]\\ F_{5}\left(x_{5}\right)=jZ_{5}v_{b}\left(\frac{\cos k_{5}l_{5}}{\sin k_{5}l_{5}}\cos k_{5}x_{5}+\sin k_{5}x_{5}\right)[\frac{Z_{3}}{Z_{4}}(\sin k_{3}l_{3}\sin k_{4}l_{4}-\cos k_{3}l_{3}\\ \;\;\;\;\;\;\;\;\;\;\;\;\;\;\;\;\;\cos k_{4}l_{4})\left(\frac{Z_{1}}{Z_{2}}\sin k_{1}l_{1}\sin k_{2}l_{2}-\cos k_{1}l_{1}\cos k_{2}l_{2}\right)-(\frac{1}{Z_{4}}\cos k_{3}l_{3}\sin k_{4}l_{4}+\\ \;\;\;\;\;\;\;\;\;\;\;\;\;\;\;\;\;\frac{1}{Z_{3}}\sin k_{3}l_{3}\cos k_{4}l_{4})(Z_{1}\sin k_{1}l_{1}\cos k_{2}l_{2}+Z_{2}\cos k_{1}l_{1}\sin k_{2}l_{2})] \end{array}$$

Substituting the coefficients of Equations (8) and (9) into the equations of Equation (7), the frequency equation on the left of the ultrasonic transducer can be written as:(12)$$\begin{array}{l} \frac{Z_{1}}{Z_{2}}\tan k_{1}l_{1}\tan k_{2}l_{2}+\frac{Z_{1}}{Z_{3}}\tan k_{1}l_{1}\tan k_{3}l_{3}+\frac{Z_{1}}{Z_{4}}\tan k_{1}l_{1}\tan k_{4}l_{4}+\\ \frac{Z_{1}}{Z_{5}}\tan k_{1}l_{1}\tan k_{5}l_{5}+\frac{Z_{2}}{Z_{3}}\tan k_{2}l_{2}\tan k_{3}l_{3}+\frac{Z_{2}}{Z_{4}}\tan k_{2}l_{2}\tan k_{4}l_{4}+\\ \frac{Z_{2}}{Z_{5}}\tan k_{2}l_{2}\tan k_{5}l_{5}+\frac{Z_{3}}{Z_{4}}\tan k_{3}l_{3}\tan k_{4}l_{4}+\frac{Z_{3}}{Z_{5}}\tan k_{3}l_{3}\tan k_{5}l_{5}+\\ \frac{Z_{4}}{Z_{5}}\tan k_{4}l_{4}\tan k_{5}l_{5}-\frac{Z_{1}}{Z_{2}}\cdot \frac{Z_{3}}{Z_{4}}\tan k_{1}l_{1}\tan k_{2}l_{2}\tan k_{3}l_{3}\tan k_{4}l_{4}-\\ \frac{Z_{1}}{Z_{2}}\cdot \frac{Z_{3}}{Z_{5}}\tan k_{1}l_{1}\tan k_{2}l_{2}\tan k_{3}l_{3}\tan k_{5}l_{5}-\frac{Z_{1}}{Z_{2}}\cdot \frac{Z_{4}}{Z_{5}}\tan k_{1}l_{1}\\ \tan k_{2}l_{2}\tan k_{4}l_{4}\tan k_{5}l_{5}-\frac{Z_{1}}{Z_{3}}\cdot \frac{Z_{4}}{Z_{5}}\tan k_{1}l_{1}\tan k_{3}l_{3}\tan k_{4}l_{4}\tan k_{5}l_{5}-\\ \frac{Z_{2}}{Z_{3}}\cdot \frac{Z_{4}}{Z_{5}}\tan k_{2}l_{2}\tan k_{3}l_{3}\tan k_{4}l_{4}\tan k_{5}l_{5}=1 \end{array}$$

There are only a rear cover plate and four piezoelectric ceramics on the left side of transducer, and the relevant parameters for piezoelectric ceramics are known. Therefore, Equation (12) can be simplified as:(13)$$\frac{Z_{1}}{Z^{\prime }}\tan k_{1}l_{1}\tan k^{\prime }l^{\prime }\left(5-4\tan^{2}k^{\prime }l^{\prime }\right)+6\tan^{2}k^{\prime }l^{\prime }-\tan^{4}k^{\prime }l^{\prime }=1$$

The right side of the section surface, AB, is the front cover plate of the longitudinal– torsional transducer and the index cylinder composite variable amplitude rod. According to the same method and results can be derived from the frequency equation, the vibration speed equation and the general solution. As a result, the frequency equation on the right side of the nodal plane is:
(14)$$\tan\left(k_{8}l_{8}\right)=\frac{k_{6}\tan k_{6}l_{6}+\beta \tan k_{6}l_{6}\tan k_{7}l_{7}-k_{7}\tan k_{7}l_{7}}{-k_{6}\tan k_{6}l_{6}\tan k_{7}l_{7}+\beta \tan k_{7}l_{7}+k_{7}}$$

### 2.2. Structural Design of Longitudinal–Torsional Transducer

As shown in Figure 1, 1 and 3 denote piezoelectric ceramics, 2 denotes electrode sheets, 4 represents the rear cover plate and 5 and 6 represent the front cover plates. The vibration velocity of the outer surface of the front cover plate is $V_{b}$, and the elastic force can be written as $F=Z_{w}V_{b}$, where $Z_{w}$ represents the input impedance. The vibration speed of the outer surface of the rear cover plate is $V_{f}$. Due to the front and rear ends of the longitudinal–torsional transducer being exposed to air, it can be considered that the elastic force is zero. The waveforms of each part under different input frequencies on both sides of the transducer node planes are shown in Figure 2. Where Ⅰ, Ⅱ indicates the wave propagation from both sides of the node, the arrow indicates the positive direction of the transducer, and the curves indicate the wave propagation schematic at different amplitudes.

In order to generate longitudinal–torsional vibration, the shape and section of the horn needs to be changed to realize the transformation of vibration modes. In this paper, a spiral groove sandwich longitudinal–torsional transducer is presented. The spiral groove structure, which can convert the longitudinal vibration into longitudinal vibration and torsional vibration, was machined at the front end of the sandwich longitudinal–torsional transducer. As a result, there will be torsional and longitudinal vibration components at the tip of tool, as shown in Figure 3. Where A-B denotes the nodal surface, C-D denotes the helix start line, *F* denotes the total force, *F_t_* denotes the torsional force, *F_l_* denotes the longitudinal force, and φ denotes the helix angle.

In Figure 3, the solid line between A-B shows the node plane in the longitudinal–torsional vibration mode, and the dashed line between C-D shows the partition interface between the longitudinal vibration part and longitudinal-torsional vibration part. At the left end of the partition interface, the piezoelectric ceramics generate axial vibration under the excitation of a sinusoidal signal generated by an ultrasonic generator, and the force is a longitudinal force, *F*. As shown in Figure 4, when the longitudinal wave moves forward along the axis of the cylinder, it will produce a transverse wave and a longitudinal wave when encountering the spiral structure. Due to the effect of the groove, the longitudinal force is separated into a longitudinal force, $F_{l}$, along the axis direction and a tangential force, $F_{t}$, perpendicular to the radius.

Figure 5 illustrates the front cover plate of the longitudinal–torsional transducer. Where C-D denotes the helix start line, *r* is the radius of the circle formed by the helix. The inner diameter and outer diameter of the front cover plate are $r_{1}$ and $r_{2}$, respectively. φ is the angle between the spiral groove and the axis of front cover plate. The length of the cylinder in front of the groove is $l_{1}$, and the length of rest is $l_{2}$. Due to the torsional vibration generated by the spiral structure, it can be considered that some sections ($l_{1}$) do not produce torsional vibration, while the rest produces not only longitudinal vibration but also torsional vibration. To simplify the analysis, it is assumed that the groove is an ideal geometric segment, i.e., the width of the groove is infinitely small.

The longitudinal force and tangential force can be determined by the following equation:(15)$$\left\{ \begin{array}{l} F_{l}=F\cdot \cos\varphi \\ F_{t}=F\cdot \sin\varphi \end{array}\right.$$
where φ is the inclination angle of the helical groove.

According to vibration theory, all tangential force components will produce a torsional movement for any section of a spiral structure, and the total torque can be expressed as:(16)$$M=\iint_{s}rfds$$
where $s=\pi \left(r_{2}{}^{2}-r_{1}{}^{2}\right)$ represents the cross-sectional area of the cylinder, *r* is the section radius at any position, *f* is the tangential force per unit area and ds=2πrdr is the element area. Therefore, the tangential force per unit area can be obtained as:(17)$$f=\frac{F\cdot \sin\left(\varphi \right)}{\pi \left(r_{2}{}^{2}-r_{1}{}^{2}\right)}$$

Equation (16) can be simplified as follows:(18)$$M=F\sin\left(\varphi \right)\cdot \frac{2\left(r_{2}{}^{3}-r_{1}{}^{3}\right)}{3\left(r_{2}{}^{2}-r^{2}{}_{1}\right)}$$

According to Equation (18), the torque, M, increases with the increases in rotation angle, φ, and notch radius, $r_{1}$, while the sectional area and polar moment of inertia decrease with the increase in notch radius, $r_{1}$. Therefore, the torsional vibration amplitude increases with the increase in rotation angle, φ, and notch radius, $r_{1}$. Due to the existence of the spiral structure, in addition to longitudinal vibration, there is also a torsional vibration in the front cover plate. Therefore, the vibration mode of the front cover plate of the transducer is a longitudinal–torsional composite vibration.

In order to avoid the influence of spiral groove parameters on the output amplitude, it was necessary to simulate the spiral structure with different parameters and select the optimal chute parameters. In this study, the single factor analysis method was used for analysis by changing the spiral parameters, such as the inclination angle, width, depth, length and distance from the small end, etc. The specific simulation results are shown in Figure 6 and Figure 7.

As shown in Figure 6, the groove depth has a great influence on the longitudinal amplitude, which changes as a quadratic function. As the groove width becomes wider, the longitudinal amplitude first decreases, then increases and finally decreases. When the groove width is 0.5 mm and 2 mm, the longitudinal amplitude reaches the maximum value. The longitudinal amplitude value fluctuates between 17 mm–20 mm. The effect of spiral groove parameters on the torsional vibration is illustrated in Figure 7. As shown in Figure 7, the influence of spiral groove structural parameters is greater than the longitudinal amplitude and natural frequency.

To sum up, the structural parameters of spiral grooves have a significant influence on the torsional amplitude. The reason for this may be that when the width, length and depth of the spiral groove change, more energy is converted into torsional vibration, which makes the longitudinal vibration decrease and torsional vibration increase. Therefore, when considering the influence of spiral groove structural parameters on the longitudinal–torsional vibration, the length and width of the spiral groove should be considered. In order to make the longitudinal–torsional composite vibration system output a larger longitudinal torsional amplitude towards the design frequency of 25 kHz, the specific parameters of the spiral groove are shown in Table 1.

The vibration modes and resonant frequencies of the longitudinal–torsional transducer were confirmed using finite element method (FEM). Figure 8 illustrates the modal velocity vector diagram of the longitudinal–torsional transducer. As shown in Figure 8, the resonance frequency of the transducer which corresponds to the longitudinal–torsional vibration is 24,609 Hz. This indicates that the longitudinal vibration that was input was successfully converted into a longitudinal–torsional composite vibration. Celaya [4] measurements with a vibrometer showed that the bending vibration mode was obtained at 20.1 kHz, with the maximum vibration amplitude obtained at 8~10 μm at the tip of the tool. In contrast to the UVAM in this paper, the maximum longitudinal vibration can reach 19 μm, and the torsional amplitude can reach 9 μm, as shown in Figure 9.

## 3. Directivity of Longitudinal–Torsional Transducer

During ultrasonic propagation, the conversion of longitudinal and transverse waves in transducers is essentially a sound field problem. Sound fields are generated by transducers, and most transducers generate a finite beam sound field. This sound field is closely related to the shape, size, vibration mode and working parameters of the transducer. Therefore, it was necessary to analyze the directivity of the transducer and evaluate the directivity of the finite beam sound field.

### 3.1. Distribution of Radiated Sound Field

The transducer source selected in this study was a piston type sound source, which generally refers to a plane shaped vibrator. When the sound source vibrates along the normal direction of the plane, the amplitude and phase of the vibration velocity at any point are the same. Based on the theory of sound field calculation [29], the sound field radiated by the transducer within a finite size can be analyzed according to the linear superposition principle. The effective radiation surface of the transducer is regarded as the combination of countless point sound sources. The sound pressure at a certain point in the radiation field is the result of the superposition of sound pressure generated at that point by all point sources. If there is a surface sound source of an arbitrary shape, the amplitude and phase of vibration at each point on the surface may be different. The sound source surface, *S*, is decomposed into an infinite number of small panels, *dS*. On each panel, *dS*, the vibration of each point can be seen as uniform, so these panels, *dS*, are considered to be point sound sources. The vibration law of the point source located in space (*x*, *y*, *z*) can be assumed to be:(19)$$u=u_{a}(x,y,z)e^{\left[\omega t-\partial (x,y,z)\right]}$$
where $u_{a}$ is the amplitude of the surface source, α is a position function, which is related to the position of the point source in space. The point source intensity, dp, can be expressed as the integral of the amplitude on the panel:(20)$$dp=j\frac{\kappa \rho_{0}c_{0}}{2\pi h(x,y,z)}dQ_{0}e^{j[\omega t-kh(x,y,z)-\alpha (x,y,z)]}$$
where h(x,y,z) is the radial propagation distance function from the center of the point source; κ represents the wave number; $\kappa =2\pi f/c_{0}$;$Q_{0}$ refers to the point source intensity and represents the volumetric velocity amplitude of the spherical source formed by the point source; and $Q_{0}=4\pi r_{0}^{2}u_{a}$.

Total sound pressure can be obtained as follows:(21)$$p=\iint_{S}j\frac{k\rho_{0}c_{0}}{2\pi h(x,y,z)}e^{j[\omega t-kh(x,y,z)-\alpha (x,y,z)]}dQ_{0}$$

Figure 10 illustrates the sound field distributions of two common piston radiation sources, where *r* is the radius of the circular piston (the rectangular piston length and width are a and b). The center of a circular piston transducer is set as coordinate origin *o*, and the plane of piston is located at the *xoy* plane. The sound field is rotationally symmetric with respect to the z axis passing through the center of the piston. Therefore, for an observation point *p*, in the sound field, the distance from the origin, *o*, is *r*, and the included angle between the position vector, *r*, and *z* axis is *θ*.

Based on the principle of point source combination, the radiated sound field of the circular piston was analyzed. The piston surface can be divided into an infinite number of small panels. The sound pressure generated at the observation point, p, by the panel with polar diameter, $r_{0}$, and polar angle, ξ, is expressed as follows:(22)$$dp=j\frac{k\rho_{0}c_{0}}{2\pi r}u_{a}dSe^{j(\omega t-kh)}$$

For circular pistons, each face element vibrates in the same phase. In order to facilitate calculations, the radiation sound pressure of the whole piston can be obtained by superimposing the face elements:(23)$$p=\iint dp=\iint_{S}j\frac{k\rho_{0}c_{0}}{2\pi r}u_{a}e^{j(\omega t-kr)}dS=j\frac{\omega \rho_{0}u_{a}}{2\pi r}e^{j(\omega t-kr)}\int_{0}^{a}\rho d\rho \int_{0}^{2\pi }e^{jk\rho \sin\theta \cos\xi }d\xi$$

Where the initial position function can be expressed as α(x,y,z)=0. For the region of *r* > *a*, the amplitude can be approximated as the distance from the piston center to the observation point, expressed as r.

Based on the Bessel function [29],
(24)$$J_{0}(x)=\frac{1}{2\pi }\int_{0}^{2\pi }e^{jx\cos\xi }d\xi ,\int xJ_{0}(x)dx=xJ_{1}(x)$$

The integral of Equation (24) can be expressed as the following equation:(25)$$p=j\omega {\frac{\rho_{0}u_{a}a}{2r}}^{2}[\frac{2J_{1}(ka\sin\theta )}{ka\sin\theta }]e^{j(\omega t-kr)}$$

The radial velocity of the mass, meanwhile, can be expressed as:(26)$$\nu_{r}=-\frac{1}{j\omega \rho_{0}}\frac{\partial p}{\partial r}=\frac{1}{\rho_{0}c_{0}}\left(1+\frac{1}{jkr}\right)p$$

As can be seen from Equation (26), in the area far away from the piston, the sound pressure decreases inversely with distance. However, the sound pressure is uneven along different directions at the same distance. Because the sound waves from different positions have different phases when reaching the observation point, the interference results in the directivity of the sound field.

### 3.2. Directivity Functions and Properties

The directivity function, i.e., the sensitivity of the directivity of the transducer, is a spatial distribution function used to describe the sound field radiated by a sound source (free far field) or the resolution and anti-interference ability of the receiver. The directivity response pattern is used to characterize the energy loss and reflect the stability during processing.

Based on the Bessel function [29],
(27)$$J_{n}(x)=\sum_{k=0}^{\infty }{\left(-1\right)}^{k}\frac{{\left(\frac{x}{2}\right)}^{2k+n}}{k!(n+k)!}$$

Therefore, the directivity of the piston sound source can be expressed as follows:(28)$$D(\theta )=\frac{{\left(p_{a}\right)}_{\theta }}{{\left(p_{a}\right)}_{\theta =0}}=\left|\frac{2J_{1}(ka\sin\theta )}{ka\sin\theta }\right|$$

It can be seen that the directivity is related to the relative ratio of the size of the piston to the wavelength. Under certain conditions in terms of the sound field environment (a certain ultrasonic frequency, f, and a constant acoustic propagation medium), it is possible to analyze the directivity by changing the radius of the piston transducer.

Figure 11 shows the directivity pattern of the piston transducer with different radii of 30 mm, 40 mm and 50 mm. As shown in Figure 11, no matter what the radius of the piston transducer is, side lobes will appear on both sides of the main lobe. As the radius of the piston transducer increases, the width of the main lobe narrows, the number of side lobes increase and the directivity of the transducer becomes stronger. With the increase in radius, the position of the maximum sidelobe moves to the middle. When θ is close to 90°, the directivity tends to be 0.

## 4. Discussion

### 4.1. Experimental Setup

During UVAM of honeycomb aramid material, the wireless power supply system for the longitudinal–torsional transducer was pre-installed on the machine spindle in order to ensure that there was no effect on the machine’s electric spindle, as shown in Figure 12. The high-frequency power of the wireless power supply system was generated by an ultrasonic generator, which was switched on and off by a control valve or control switch, and the interface was generally guaranteed for three to four years of service life. The force date was recorded by a three-way force sensor, which was mounted on the acrylic fixing plate of the worktable.

It is difficult to measure the cutting force in x, y and z directions simultaneously, so a three-way force sensor was arranged on the acrylic plate. First, the honeycomb aramid material was fixed on the acrylic plate, and when the disc cutter cut the material, the three component forces generated were transmitted to the three-way force sensor through the acrylic plate. The sensor amplified the received signals through a charge amplifier, stored them in a data acquisition card, and finally presented the data through a data acquisition device (computer). Figure 13 shows a schematic diagram of the cutting force system. The test field installation is shown in Figure 14.

### 4.2. Cutting Force

Compared with traditional milling, when the spindle speed is 1500 r/min and the cutting depth is 2 mm, the cutting forces along three directions are reduced by approximately 40–50%. The measurement results for the cutting force during UVAM are shown in Figure 15.

### 4.3. Directivity Evaluation

Concerning machining in the context of UVAM, machining stability can be expressed in terms of the maximum depth of the cut produced by the milling tool at different cutting speeds. The maximum axial cutting depth under different processing methods when machining honeycomb aramid material with a length, width and thickness of 100 mm, 100 mm and 12 mm, respectively, is illustrated in Figure 12. As shown in Figure 16, when the spindle speed is low, the maximum cutting depth during UVAM is higher than that of traditional machining, and the directivity of the transducer directivity function is also stronger, which indicates that UVAM can significantly improve the stability of the milling process.

## 5. Conclusions

This paper has proposed a longitudinal–torsional transducer for honeycomb aramid material during ultrasonic vibration assisted milling (UVAM). The directivity of transducer was analyzed via a simulation and an experiment, providing a new evidence for the processing of honeycomb aramid material and wide application. The major conclusions were listed as the following:The longitudinal wave can be converted to a transverse wave via the longitudinal–torsional transducer, thus realizing longitudinal–torsional conversion.During UVAM, the maximum longitudinal vibration can reach 19 μm, and the maximum torsional amplitude at the top of the disk milling cutter can reach 9 μm.Compared with traditional milling, the cutting force along three directions during UVAM with a longitudinal–torsional transducer is significantly reduced. Moreover, the directivity is stronger and the cutting depth is greater under the same cutting speed. This shows that UVAM technology can improve the stability of the milling process.

## Figures and Tables

**Figure 1 micromachines-13-02154-f001:**
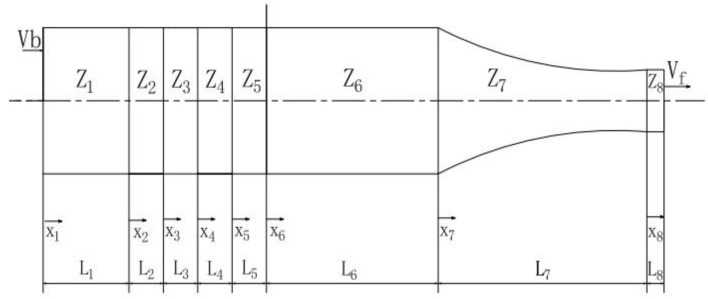
Structural diagram of longitudinal–torsional transducer.

**Figure 2 micromachines-13-02154-f002:**
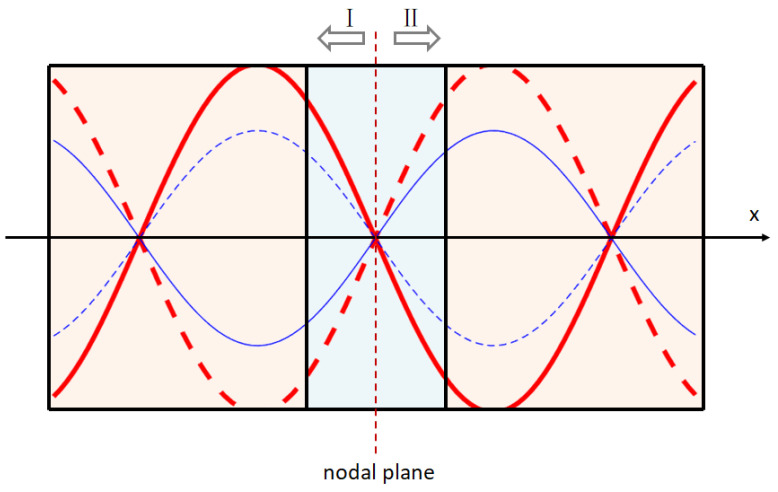
Waveform diagram of nodal plane.

**Figure 3 micromachines-13-02154-f003:**
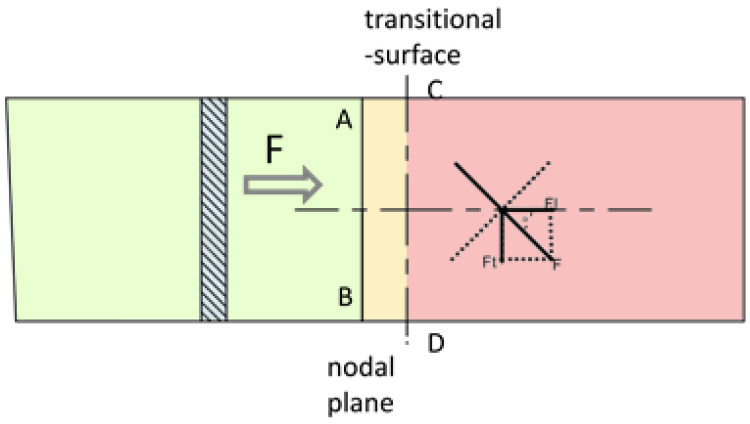
Schematic diagram of vibration propagation.

**Figure 4 micromachines-13-02154-f004:**
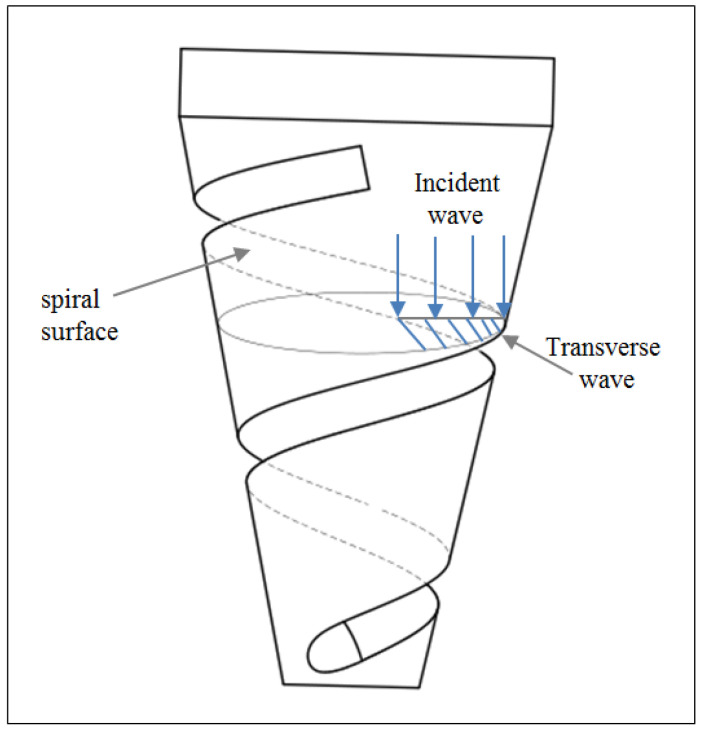
Schematic diagram of spiral structure.

**Figure 5 micromachines-13-02154-f005:**
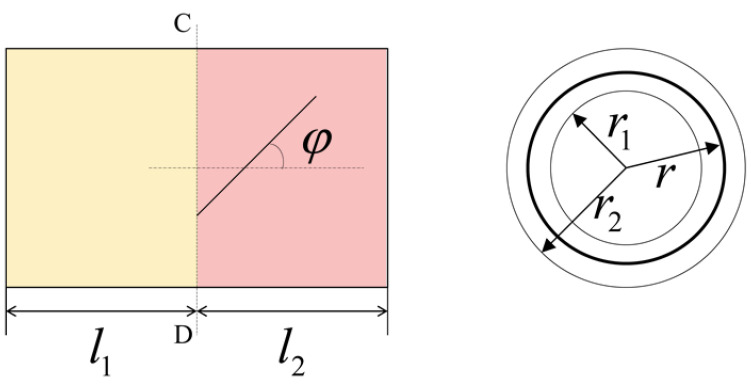
Front cover plate of longitudinal–torsional transducer.

**Figure 6 micromachines-13-02154-f006:**
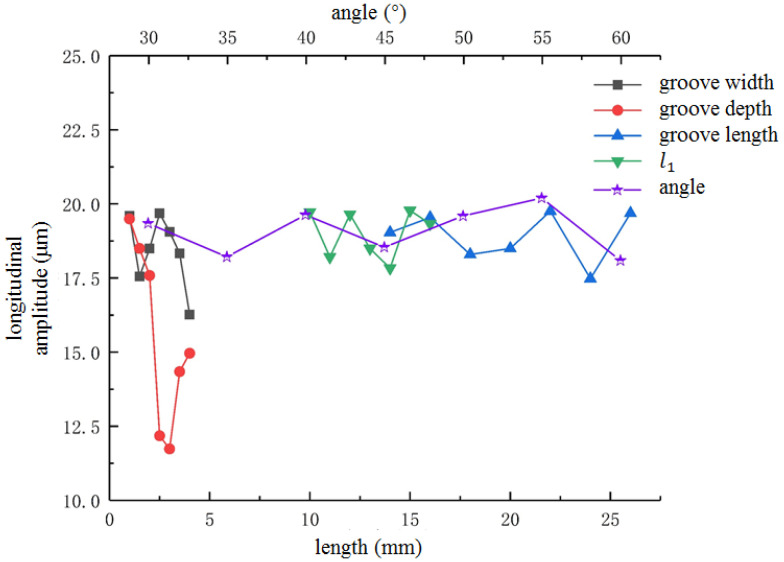
Effect of spiral groove parameters on longitudinal vibration.

**Figure 7 micromachines-13-02154-f007:**
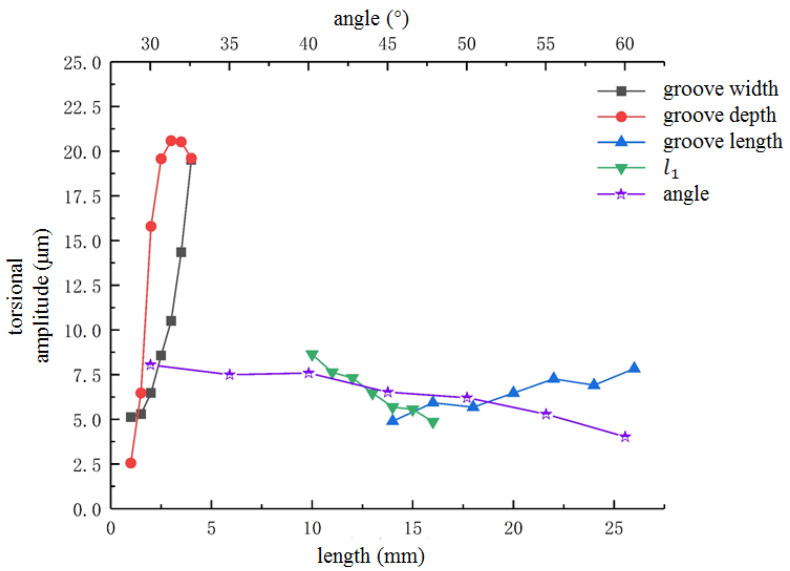
Effect of spiral groove parameters on torsional vibration.

**Figure 8 micromachines-13-02154-f008:**
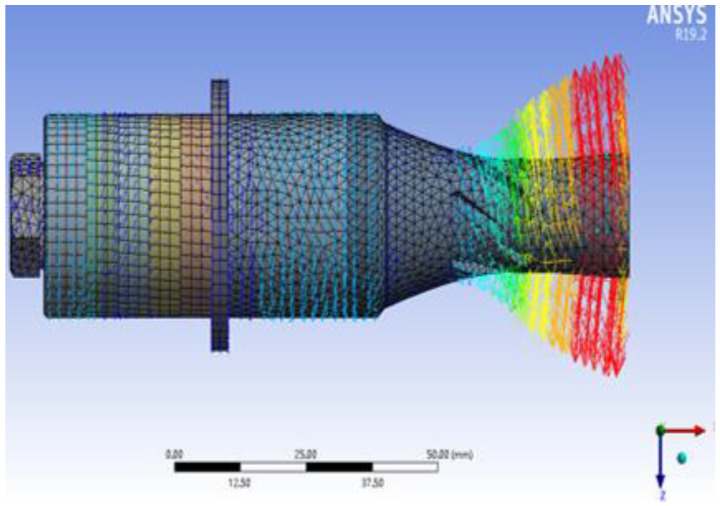
Modal velocity vector diagram of longitudinal–torsional transducer.

**Figure 9 micromachines-13-02154-f009:**
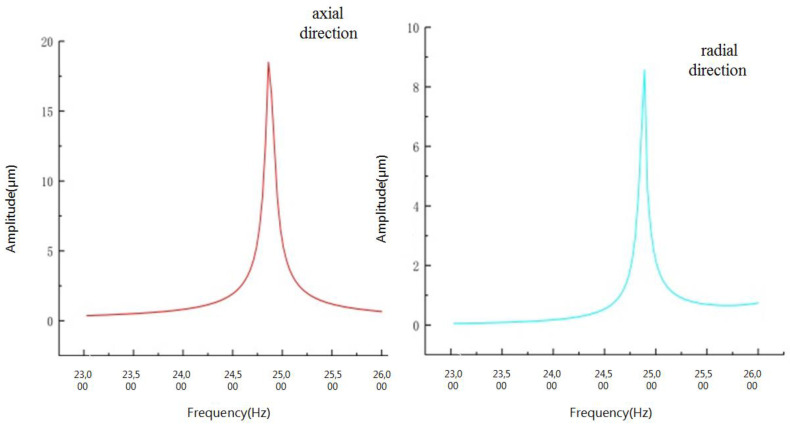
Longitudinal–torsional amplitude.

**Figure 10 micromachines-13-02154-f010:**
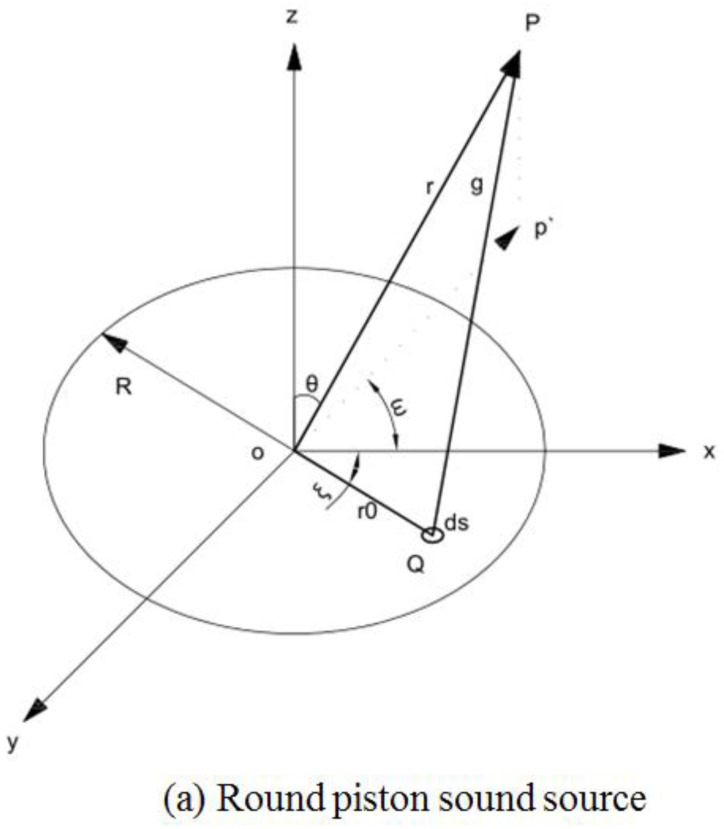
Distribution of radiated sound field. (**a**) round piston sound source, (**b**) rectangular piston sound source.

**Figure 11 micromachines-13-02154-f011:**
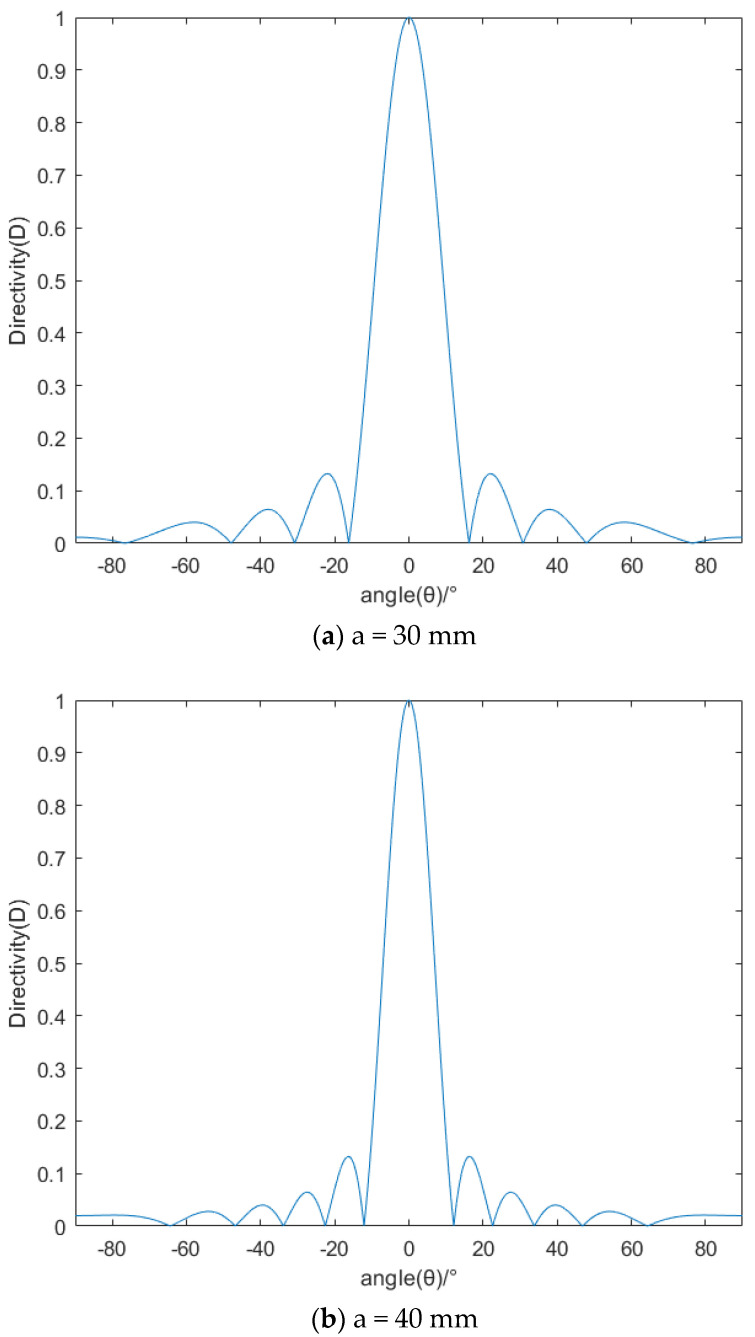
Directivity pattern of the piston transducer with different radii of 30 mm (**a**), 40 mm (**b**) and 50 mm (**c**).

**Figure 12 micromachines-13-02154-f012:**
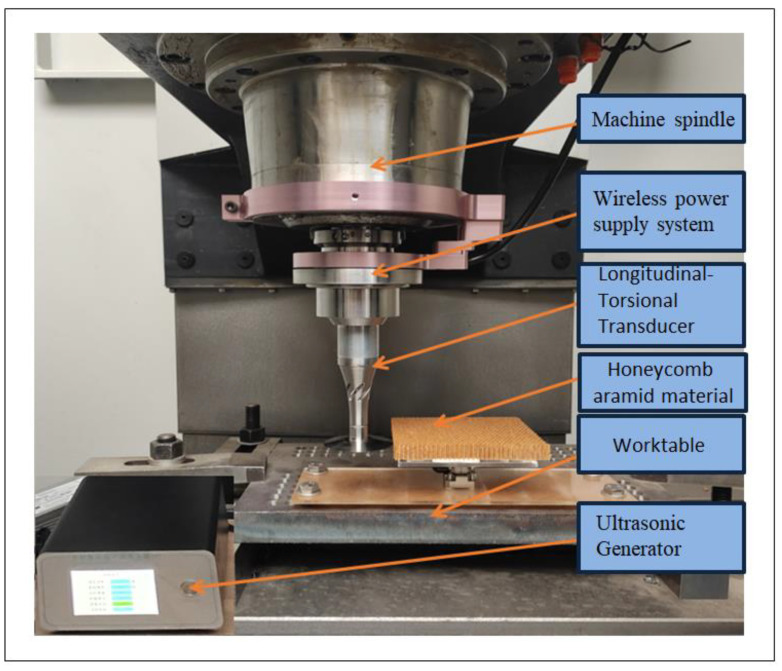
Experimental setup.

**Figure 13 micromachines-13-02154-f013:**
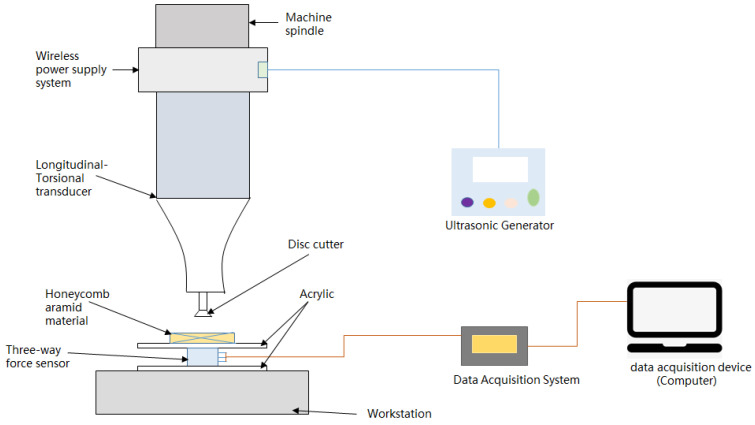
Schematic diagram of the cutting force system.

**Figure 14 micromachines-13-02154-f014:**
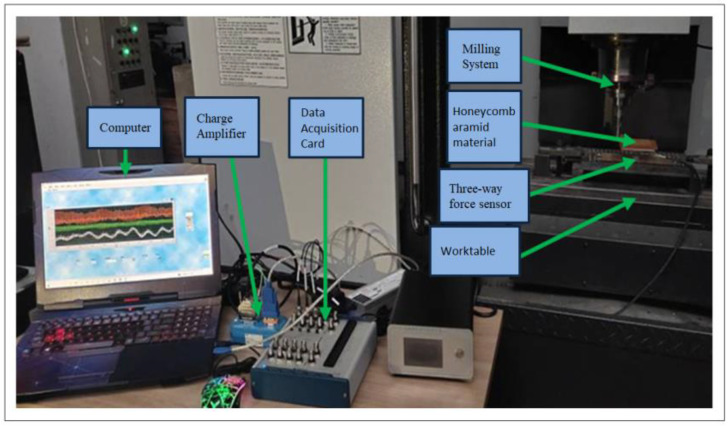
System for measurement cutting force.

**Figure 15 micromachines-13-02154-f015:**
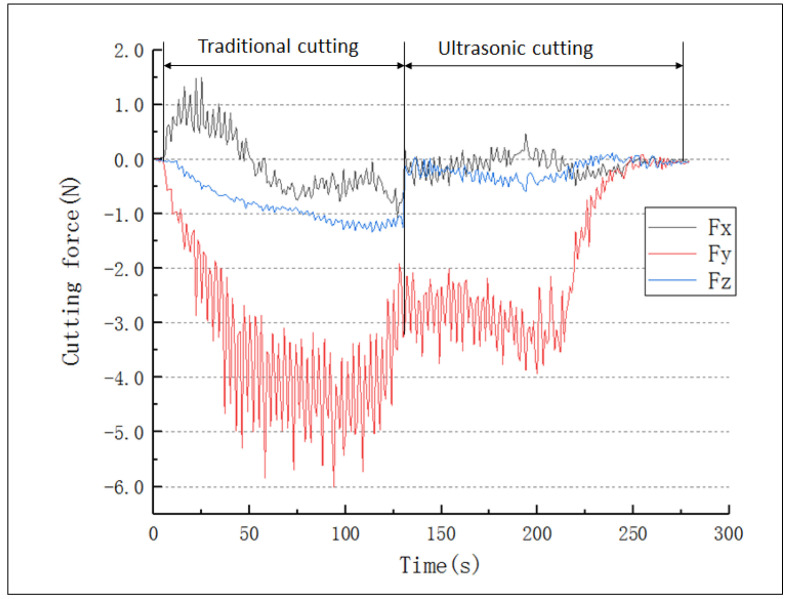
Cutting force under different processing methods.

**Figure 16 micromachines-13-02154-f016:**
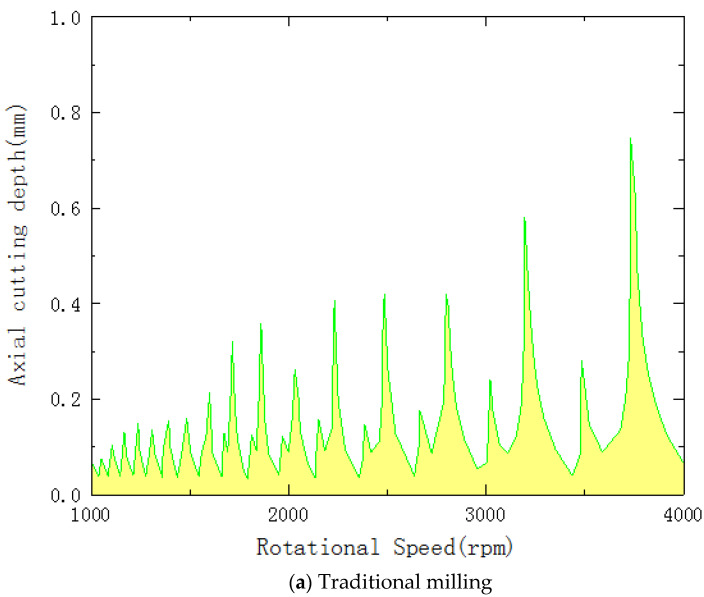
Directivity pattern under different processing methods.

**Table 1 micromachines-13-02154-t001:** Specific parameters of transducer structure.

Number of Grooves	Length	Depth	Width	*l* _1_	Angle
6	22 mm	1.5 mm	2 mm	10 mm	45°

## Data Availability

Not applicable.

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
