# Peer review of "Design of Longitudinal–Torsional Transducer and Directivity Analysis during Ultrasonic Vibration-Assisted Milling of Honeycomb Aramid Material"

_micromachines, 2022, doi:10.3390/mi13122154_

Round 1

Reviewer 1 Report

1.       The meaning is not clear: “machining stability can be expressed by the maximum cutting depth under different cutting speeds”, Celaya´s works were missed where She/he gave the fundamentals of US assisted processes.

2.       Works like International Journal of Materials and Product Technology 37 (1-2), 60-70 DOI10.1504/IJMPT.2010.029459

3.       The booster by you is similar to Celaya´s. other authors gave other values, see Effects of ultrasonics-assisted face milling on surface integrity and fatigue life of Ni-Alloy 718, Journal of Materials Engineering and Performance 25 (11), 5076-5086

4.       UVAM, the maximum longitudinal vibration can reach 19 um. Compare your amplification with above ones.

Duration of the wireless power system, how much is the life of this interface?

Figure 4 is quite new, a pity you wrote a poor state of the art to define exactly the novelty.

Please submit another version soon.

Author Response

Dear reviewer,

We thank you very much for giving us an opportunity to revise our manuscript, we appreciate editor and reviewer very much for their positive and constructive comments and suggestions on our manuscript entitled “Design of Longitudinal-torsional Transducer and Directivity Analysis during Ultrasonic Vibration Assisted Milling of Honeycomb Aramid Material”. Those comments are all valuable and helpful for revising and improving our paper. We have studied comments carefully and have made revisions which marked in red in the paper. The main corrections in the paper and the responds to the comments are as flowing:

Reviewer #1:

  1. The meaning is not clear: “machining stability can be expressed by the maximum cutting depth under different cutting speeds”, Celaya´s works were missed where She/he gave the fundamentals of US assisted processes.

Response: We appreciate your precious suggestions. In view of the above problems, we have modified part of the introduction, and revised part of the discussion to illustrate these problems.

To respond to the suggestions, Firstly, The cited literature describes the implementation of a longitudinal torsional transducer, which illustrates the limitations of conventional processing methods for honeycomb aramid materials. Next, two implementations of the longitudinal twist transducer are studied, and the research results and differences between the two implementations by different scholars are compared. Then, we analyze the advantages of this form and the process complexity of combining axial and tangential piezoelectric ceramics based on the transformation of vibration modes. Finally, we choose the mode-conversion approach.

We re-narrated the description of the stability of the milling process to make it sufficiently specific. In contrast to Celaya's paper, where she focused on the surface roughness of machined alloy steel, but the problems of shutdown looseness and chattering caused by insufficient stiffness that occurred during the experiment were not addressed, this paper focuses on characterizing UVAM to improve machining stability through stability leaflet plots.

On page 1: In the process of vibration mode conversion, torsional vibrations in transmission rod can be obtained based on the coupling and degradation of horn form longitudinal vibration. Patrick Harkness [3] proposed two design methods for longitudinal-torsional mode transitions. Celaya[4] proposed booster can improve surface quality but with low amplitude. Liu [5] demonstrated two forms of thread grooves which can degrade longitudinal vibrations into longitudinal-torsional vibrations. Yahya [6], Gao [7], Yuan [8], Zhang [9], and Wu [10] developed the L-T transducers based on theoretical and simulation analysis, and applied it to the processing of Ti-6Al-4V. Budairi[11] used numerical, analytical and experimental methods to evaluate the transducer performance under different excitations based on the L-T response. In addition, by setting different degradation modes on the cylinder surface, it can be applied to ultrasonic motors [12-14]. Shen[15] conducted an assisted ultrasonic vibration micro-milling process on aluminum alloy material and used a combination of machining parameters to demonstrate that ultrasonic vibration has a significant improvement on surface roughness; Chen[16] came to the same conclusion for silicon carbide ceramic grinding.Cleary [17] and Wang [18] proposed an L-T ultrasound needle device for medical treatment and implantation applications, which provides a potential application scenario for the promotion of L-T mode. NumanoÄŸlu [19] and AkbaÅŸ [20] solved and derived the vibrations of composite rods and beams for different boundary conditions using different theoretical methods, providing ideas for the theoretical derivation of L-T transducer. However, in the processing of honeycomb aramid materials by UVAM, there is fewer research on longitudinal-torsional transducer, which limits the improvement of the machinability of honeycomb aramid materials.

On page 2: In addition, the stability lobes diagram is of great significance for the optimization of equipment structure and process parameters, which is mainly affected by the directivity of transducer during UVAM. At present, the relevant research mainly focuses on analysis methods of stability [24], correctness of the stability lobes diagram [25], and increased equivalent stiffness by filling the honeycomb material with viscoelastic damping material [26]. Suárez [27] showed that the application of ultrasonic milling can improve the form of tool wear patter and make the evolution more stable. However, the research on directivity of transducer during ultrasonic vibration assisted milling (UVAM) of honeycomb aramid material is relatively few.

On page 19: In machining of UVAM, machining stability can be expressed in terms of the max-imum depth of cut produced by the milling tool at different cutting speeds. The maxi-mum axial cutting depth under different processing methods when machining honey-comb aramid material with length, width and thickness of 100mm, 100mm and 12mm are illustrated in Figure 15. As shown in Figure 15, when the spindle speed is low, the maximum cutting depth of UVAM is higher than that of traditional machining, and the directivity of transducer directivity function is also stronger, which indicates that UVAM can significantly improve the stability of milling process.

  1. Works like International Journal of Materials and Product Technology 37 (1-2), 60-70 DOI: 10.1504/IJMPT.2010.029459

  1. The booster by you is similar to Celaya´s. other authors gave other values, see Effects of ultrasonics-assisted face milling on surface integrity and fatigue life of Ni-Alloy 718, Journal of Materials Engineering and Performance 25 (11), 5076-5086

  1. UVAM, the maximum longitudinal vibration can reach 19 um. Compare your amplification with above ones.
    Response:Thank you very much for your valuable comments. In response to this suggestion, we have added some references and revised the introduction to help clarify the problem. First, the UVAM method outperforms conventional machining methods for machining hard-to-process materials by introducing an additional ultrasonic vibration excitation to the conventional machine tool. This allows the machine tool to rotate while machining while adding an ultrasonic vibration to the tool head in one or more directions, thereby improving the machining efficiency and surface finish of the original machine tool equipment. Secondly, due to the above reasons, many scholars have applied the longitudinal torsional vibration to the field of material processing and proved that this method can improve the surface quality, machining force, fatigue performance, etc. Therefore, we conducted the present research work.

By comparing Alfredo Sua´rez paper, the paper focuses on the analysis of UA on Ni-Alloy 718 material surface roughness, machining force, surface hardness, surface topography, tool damage and fatigue test performance. Comparing the machining forces, the cutting forces in x, y and z directions are smaller in this paper, and the magnitude of milling force reduction is more obvious when the conventional milling and UVAM milling forces are established in the same coordinate system.Celaya's measurement results with a vibrometer show that the bending vibration mode is obtained at 20.1 kHz with the largest vibration amplitude at 8 ~ 10 μ m at the tool tip. In contrast to the UVAM in this paper, the maximum longitudinal vibration can reach 19 um.

On page 1: In the process of vibration mode conversion, torsional vibrations in transmission rod can be obtained based on the coupling and degradation of horn form longitudinal vibration. Patrick Harkness [3] proposed two design methods for longitudinal-torsional mode transitions. Celaya[4] proposed booster can improve surface quality but with low amplitude. Liu [5] demonstrated two forms of thread grooves which can degrade longitudinal vibrations into longitudinal-torsional vibrations. Yahya [6], Gao [7], Yuan [8], Zhang [9], and Wu [10] developed the L-T transducers based on theoretical and simulation analysis, and applied it to the processing of Ti-6Al-4V. Budairi[11] used numerical, analytical and experimental methods to evaluate the transducer performance under different excitations based on the L-T response. In addition, by setting different degradation modes on the cylinder surface, it can be applied to ultrasonic motors [12-14]. Shen[15] conducted an assisted ultrasonic vibration micro-milling process on aluminum alloy material and used a combination of machining parameters to demonstrate that ultrasonic vibration has a significant improvement on surface roughness; Chen[16] came to the same conclusion for silicon carbide ceramic grinding.Cleary [17] and Wang [18] proposed an L-T ultrasound needle device for medical treatment and implantation applications, which provides a potential application scenario for the promotion of L-T mode. NumanoÄŸlu [19] and AkbaÅŸ [20] solved and derived the vibrations of composite rods and beams for different boundary conditions using different theoretical methods, providing ideas for the theoretical derivation of L-T transducer. However, in the processing of honeycomb aramid materials by UVAM, there is fewer research on longitudinal-torsional transducer, which limits the improvement of the machinability of honeycomb aramid materials.

On page 2: In addition, the stability lobes diagram is of great significance for the optimization of equipment structure and process parameters, which is mainly affected by the directivity of transducer during UVAM. At present, the relevant research mainly focuses on analysis methods of stability [24], correctness of the stability lobes diagram [25], and increased equivalent stiffness by filling the honeycomb material with viscoelastic damping material [26]. Suárez [27] showed that the application of ultrasonic milling can improve the form of tool wear patter and make the evolution more stable. However, the research on directivity of transducer during ultrasonic vibration assisted milling (UVAM) of honeycomb aramid material is relatively few.

On page 11: The vibration modes and resonant frequencies of longitudinal-torsional transducer could be confirmed using finite element method (FEM). Figure 8 illustrates the modal velocity vector diagram of longitudinal-torsional transducer. As shown in Figure 8, the resonance frequency of transducer corresponding to longitudinal-torsional vibration is 24609 Hz. This indicates that the input longitudinal vibration has been successfully converted into longitudinal-torsional composite vibration. Celaya[4] measurements with a vibrometer showed that the bending vibration mode was obtained at 20.1 kHz with the maximum vibration amplitude at 8 ~ 10 μ m at the tip of the tool. In contrast to the UVAM in this paper, the maximum longitudinal vibration can reach 19 um and the torsional amplitude can reach 9 um, which can be shown in Figure 9.

Duration of the wireless power system, how much is the life of this interface?

Response: Thank you very much for your precious suggestion. our interpretation of the problem is not clear enough, so we added several parts to address this issue in Section 4. Firstly, we introduce the issue of pre-installation of the wireless power supply system on the machine spindle. Secondly, we explain the high-frequency excitation of the wireless power supply system and its control, and illustrate its durability and interface lifetime.

On page 17: During UVAM of honeycomb aramid material, the wireless power supply system for the longitudinal-torsional transducer was pre-installed on the machine spindle in order to ensure that there was no effect on the machine's electric spindle, as shown in Figure 12. The high-frequency power of the wireless power supply system was generated by ultrasonic generator, which was switched on and off by a control valve or control switch, and the interface was generally guaranteed for three to four years of service life. The force date was recorded by a three-way force sensor, which is mounted on the acrylic fixing plate of the worktable. The cutting force measurement system is shown in Figure 13.

Figure 4 is quite new, a pity you wrote a poor state of the art to define exactly the novelty.

Response: We appreciate your kind suggestions. We apologize for this failure in the previous version. To solve this problem, We have changed the perspective of Figure 4 and redrawn it to better show the incident wave, the transverse wave refraction occurring on the surface of the spiral groove, and to better explain the conversion of the longitudinal vibration into a longitudinal-torsional composite vibration.

Figure 4. Schematic diagram of spiral structure.

Reviewer 2 Report

In this study, a longitudinal-torsional transducer model for ultrasonic vibration assisted milling (UVAM) of honeycomb aramid is investigated. The transducer model is considered as a sandwich piezoelectric ceramic medium. The directivity of transducer model is studied by using the longitudinal-torsional transformation theory. Variations of parameters such as spiral groove parameter, modal speed, amplitude, directivity pattern, cutting force are examined for the transducer under UVAM. Also, the performance of transducer is verified by experimental studies.

The issue of the study is timely, interesting, and well-organized. Language is comprehensible. After the following points are examined by the authors, the manuscript can be accepted for publication:

1.      Some shapes need to be enhanced in resolution and size: Figures 4, 8, 9, 10, 11 (Especially Fig. 11), and 15.

2.      Although seven regions are seen in the transducer example, why are five displacement and stress boundary conditions given in Eq. (7)? The authors are invited to comment on this point.

3.      In Eqs. (8) and (9), Ai and Bi (i = 1, … , 5) coefficients are obtained, but Eqs. (5) and (6) have A and B. So why are not Eqs. (5) and (6) given in terms of Ai and Bi? The authors are invited to comment on this point. What are Zi coefficients seen in Eqs. (8)-(12)? This should be mentioned.

4.      Figure sequence should be checked. There are two Figures 7 on page 7.

5.      How is the longitudinal and tangential force determined on the transducer? A schematic representation should be included.

6.      For example, Q0 is not defined in Eq. (19). All equations should be checked and there should be no undefined parameters. Is the integral in Eq. (20) incomplete? There is no differential multiplier at the end of integral? dS?

7.      The references in the introduction are appropriate but inadequate. Therefore, the introduction should be expanded by adding the following articles to the introduction:

-        A study of surface roughness variation in ultrasonic vibration-assisted milling. The International Journal of Advanced Manufacturing Technology volume 58, 553–561 (2012)

-        On dynamic analysis of nanorods, International Journal of Engineering Science 130, 33-50, 2018.

-        Dynamic Analysis of a Fiber-Reinforced Composite Beam under a Moving Load by the Ritz Method. Mathematics 2021, 9, 1048. https://doi.org/10.3390/math9091048

-        A design approach for longitudinal–torsional ultrasonic transducers. Sensors and Actuators A: Physical. 198, 2013, 99-106

-        Design and development of longitudinal and torsional ultrasonic vibration-assisted needle insertion device for medical applications. Computer-Aided Design and Applications, 19(4), 797–811; 2022

-        The effect of torsional vibration in longitudinal–torsional coupled ultrasonic vibration-assisted grinding of silicon carbide ceramics. Materials (Basel). 2021 Feb 2;14(3):688. doi: 10.3390/ma14030688

8.      The reference list is not appropriate in terms of journal abbreviation (ISO 4) and line spacing. The reference list should be checked.

9.      What can be the contribution of proposed longitudinal-torsional transducer model to the application in mechanical engineering or other disciplines? In is expected that the authors briefly emphasize this point in the Conclusions section.

Author Response

Dear reviewer,

We thank you very much for giving us an opportunity to revise our manuscript, we appreciate editor and reviewer very much for their positive and constructive comments and suggestions on our manuscript entitled “Design of Longitudinal-torsional Transducer and Directivity Analysis during Ultrasonic Vibration Assisted Milling of Honeycomb Aramid Material”. Those comments are all valuable and helpful for revising and improving our paper. We have studied comments carefully and have made revisions which marked in red in the paper. The main corrections in the paper and the responds to the comments are as flowing:

Reviewer #2:

In this study, a longitudinal-torsional transducer model for ultrasonic vibration assisted milling (UVAM) of honeycomb aramid is investigated. The transducer model is considered as a sandwich piezoelectric ceramic medium. The directivity of transducer model is studied by using the longitudinal-torsional transformation theory. Variations of parameters such as spiral groove parameter, modal speed, amplitude, directivity pattern, cutting force are examined for the transducer under UVAM. Also, the performance of transducer is verified by experimental studies.

The issue of the study is timely, interesting, and well-organized. Language is comprehensible.

general reply: Thanks very much for taking your time to review this manuscript. I really appreciate all your comments and suggestions! Please find my itemized responses in below and my revisions/corrections in the re-submitted files.

  1. Some shapes need to be enhanced in resolution and size: Figures 4, 8, 9, 10, 11 (Especially Fig. 11), and 15.

Response: Thanks a lot for your opinion. According to your suggestion, we redrew Figure 4, adjusted the resolution of Figure 8 and Figure 9, split Figure 11 and Figure 15, and resized and rearranged the images. You can find our revised entry in the resubmitted revision with the revised images as follows:

Figure 4. Schematic diagram of spiral structure.

Figure 8. Modal velocity vector diagram of longitudinal-torsional transducer.

Figure 9. Longitudinal - torsional amplitude.

Figure 10. Distribution of radiated sound field.

  • a=30mm
  • a=40mm
  • a=40mm

Figure 11. Directivity pattern of piston transducer.

  • Traditional Milling

  • Ultrasonic Milling

Figure 15. Directivity pattern under different processing methods.

  1. Although seven regions are seen in the transducer example, why are five displacement and stress boundary conditions given in Eq. (7)? The authors are invited to comment on this point.

Response: Thank you very much for your suggestions on this article. Equation 7 is a description of the boundary conditions on the left side of the AB section face of the longitudinal torsional transducer. On the left side of the AB section face, it is divided into five parts based on the difference in structure, and the boundary conditions of each part are derived. Following the same idea and method, the boundary conditions, frequency equation and vibration velocity equation of the three parts on the right side of the AB section face can be solved. In order to save space, the derivation process is not reflected in this work, but we add a description of the method in Section 2.1. Also, to better facilitate your understanding, we have attached the derivation process for the right-hand side of the nodal plane.

On page 7: The right side of the section surface AB is the front cover plate of the longitudinal torsional transducer and the index cylinder-composite variable amplitude rod, according to the same method and ideas can be derived from the frequency equation, the vibration speed equation and the general solution. As a result, the frequency equation on the right side of the nodal plane is:

                     ï¼ˆ14)

Appendix:

The frequency equation and the vibration speed equation on the right side of the nodal plane AB is:

                                                                 (1)

                                                                 (2)

The distribution of stresses is:

                                              (3)

The boundary conditions on the right side of the nodal plane is:

                                                                                                (4)

Similarly, the values of the pending coefficients A6, A7, A8, B6, B7, B8 can be found by substituting the boundary conditions (4) on the right-hand side into the vibration velocity equation (2) and the stress distribution equation (3).

                           (5)

Substituting the coefficients of (5) into (4), the frequency equation on the right side of the nodal plane is:

                    (6)

  1. In Eqs. (8) and (9), Ai and Bi (i = 1, … , 5) coefficients are obtained, but Eqs. (5) and (6) have A and B. So why are not Eqs. (5) and (6) given in terms of Ai and Bi? The authors are invited to comment on this point. What are Zi coefficients seen in Eqs. (8)-(12)? This should be mentioned.

Response: Thank you very much for your comments. Equation 5 and Equation 6 are expressions that give a general solution to the vibration equation for an arbitrary cross section and solve for its corresponding stress expression, which has wide generality and practicality. Equation 8 and Equation 9 are based on Equation 5 and Equation 6 for the detailed expression of the equation of state of each part of the left side of the transducer section AB, so it carries the coefficients (Ai, Bi). Meanwhile, the meaning of Zi is explained

On page 5: Where,  is the characteristic acoustic impedance of each part of the material on the left side of the nodal plane,   .

  1. Figure sequence should be checked. There are two Figures 7 on page 7.

Response: Thank you very much for your valuable comments. We are very sorry for this mistake. The figures in the paper have been carefully checked and proofread.

On page 8: Figure 3

Figure 3. Schematic diagram of vibration propagation.

  1. How is the longitudinal and tangential force determined on the transducer? A schematic representation should be included.

Response: Thank you very much for your valuable comments. Thank you very much for your valuable comments. We have added a schematic of the three-way force measurement system and redescribed the measurement principle, which you can see on page 18.

On page 18:It is difficult to measure the cutting force in x, y and z directions simultaneously, so three-way force sensor is arranged on the acrylic plate. The honeycomb aramid material is fixed on the first acrylic plate, and when the disc cutter cuts the material, the three component forces generated are transmitted to the three-way force sensor through the acrylic plate. The sensor amplifies the received signals through a charge amplifier, stores them in a data acquisition card, and finally presents the data through a data acquisition device (computer). Figure 13 shows a schematic diagram of the cutting force system. The test field installation is shown in Figure 14.

Figure 13. Schematic diagram of the cutting force system.

  1. For example, Q0 is not defined in Eq. (19). All equations should be checked and there should be no undefined parameters. Is the integral in Eq. (20) incomplete? There is no differential multiplier at the end of integral? dS?

Response: Thank you very much for your valuable comments. We have carefully checked and revised every equation in the paper and explained the quoted letters thoroughly, as you can see in the revised version.

On page 13: where,  is the radial propagation distance function from the center of point source;  represents the wave number,; is called the point source intensity and represents the volumetric velocity amplitude of the spherical source formed by the point source, .

Total sound pressure can be obtained as follows:

 (21)

  1. The references in the introduction are appropriate but inadequate. Therefore, the introduction should be expanded by adding the following articles to the introduction:

-        A study of surface roughness variation in ultrasonic vibration-assisted milling. The International Journal of Advanced Manufacturing Technology volume 58, 553–561 (2012)

-        On dynamic analysis of nanorods, International Journal of Engineering Science 130, 33-50, 2018.

-        Dynamic Analysis of a Fiber-Reinforced Composite Beam under a Moving Load by the Ritz Method. Mathematics 2021, 9, 1048. https://doi.org/10.3390/math9091048

-        A design approach for longitudinal–torsional ultrasonic transducers. Sensors and Actuators A: Physical. 198, 2013, 99-106

-        Design and development of longitudinal and torsional ultrasonic vibration-assisted needle insertion device for medical applications. Computer-Aided Design and Applications, 19(4), 797–811; 2022

-        The effect of torsional vibration in longitudinal–torsional coupled ultrasonic vibration-assisted grinding of silicon carbide ceramics. Materials (Basel). 2021 Feb 2;14(3):688. doi: 10.3390/ma14030688

Response: Thank you for your introduction to these wonderful research work. According to your suggestion, we properly cite these articles as:

  1. Celaya, A.; Lopez de Lacalle, L. N.;  Campa, F. J.; Lamikiz, A., Ultrasonic Assisted Turning of mild steels. International journal of materials & product technology 2010, 37 (1), 60.
  2. Al-Budairi, H.; Lucas, M.; Harkness, P., A design approach for longitudinal–torsional ultrasonic transducers. Sensors and Actuators A: Physical 2013, 198, 99-106.
  3. Shen, X.-H.; Zhang, J.;  Xing, D. X.; Zhao, Y., A study of surface roughness variation in ultrasonic vibration-assisted milling. The International Journal of Advanced Manufacturing Technology 2012, 58 (5), 553-561.
  4. Chen, Y.; Su, H.;  He, J.;  Qian, N.;  Gu, J.;  Xu, J.; Ding, K., The Effect of Torsional Vibration in Longitudinal–Torsional Coupled Ultrasonic Vibration-Assisted Grinding of Silicon Carbide Ceramics. Materials 2021, 14 (3), 688.
  5. Cleary, R.; Wallace, R.;  Simpson, H.;  Kontorinis, G.; Lucas, M., A longitudinal-torsional mode ultrasonic needle for deep penetration into bone. Ultrasonics 2022, 106756.
  6. Wang, Y.; Cai, Y.; Lee, Y.-S., Design and Development of Longitudinal and Torsional Ultrasonic Vibration-Assisted Needle Insertion Device for Medical Applications. Computer-Aided Design and Application 2021, 19 (4), 797-811.
  7. NumanoÄŸlu, H. M.; Akgöz, B.; Civalek, Ö., On dynamic analysis of nanorods. International Journal of Engineering Science 2018, 130, 33-50.
  8. AkbaÅŸ, Åž. D.; Ersoy, H.;  Akgöz, B.; Civalek, Ö., Dynamic analysis of a fiber-reinforced composite beam under a moving load by the Ritz method. Mathematics 2021, 9 (9), 1048.
  9. Suárez, A.; Veiga, F.; de Lacalle, L. N. L.;  Polvorosa, R.;  Lutze, S.; Wretland, A., Effects of ultrasonics-assisted face milling on surface integrity and fatigue life of Ni-Alloy 718. Journal of Materials Engineering and Performance 2016, 25 (11), 5076-5086.

On page 13: In the process of vibration mode conversion, torsional vibrations in transmission rod can be obtained based on the coupling and degradation of horn form longitudinal vibration. Patrick Harkness [3] proposed two design methods for longitudinal-torsional mode transitions.Celaya[4] proposed booster can improve surface quality but with low amplitude. Liu [5] demonstrated two forms of thread grooves which can degrade longitudinal vibrations into longitudinal-torsional vibrations. Yahya [6], Gao [7], Yuan [8], Zhang [9], and Wu [10] developed the L-T transducers based on theoretical and simulation analysis, and applied it to the processing of Ti-6Al-4V. Budairi[11] used numerical, analytical and experimental methods to evaluate the transducer performance under different excitations based on the L-T response. In addition, by setting different degradation modes on the cylinder surface, it can be applied to ultrasonic motors [12-14]. Shen[15] conducted an assisted ultrasonic vibration micro-milling process on aluminum alloy material and used a combination of machining parameters to demonstrate that ultrasonic vibration has a significant improvement on surface roughness; Chen[16] came to the same conclusion for silicon carbide ceramic grinding.Cleary [17] and Wang [18] proposed an L-T ultrasound needle device for medical treatment and implantation applications, which provides a potential application scenario for the promotion of L-T mode. NumanoÄŸlu [19] and AkbaÅŸ [20] solved and derived the vibrations of composite rods and beams for different boundary conditions using different theoretical methods, providing ideas for the theoretical derivation of L-T transducer. However, in the processing of honeycomb aramid materials by UVAM, there is fewer research on longitu-dinal-torsional transducer, which limits the improvement of the machinability of hon-eycomb aramid materials.

On page 2: In addition, the stability lobes diagram is of great significance for the optimization of equipment structure and process parameters, which is mainly affected by the directivity of transducer during UVAM. At present, the relevant research mainly focuses on analysis methods of stability [24], correctness of the stability lobes diagram [25], and increased equivalent stiffness by filling the honeycomb material with viscoelastic damping material [26]. Suárez [27] showed that the application of ultrasonic milling can improve the form of tool wear patter and make the evolution more stable. However, the research on directivity of transducer during ultrasonic vibration assisted milling (UVAM) of honeycomb aramid material is relatively few.

  1. The reference list is not appropriate in terms of journal abbreviation (ISO 4) and line spacing. The reference list should be checked.

Response: Thank you very much for your precious comments. Based on the author guidelines, we will invite the editors to make appropriate adjustments to the references.

Figure . Quick Reference Formatting Guide

  1. What can be the contribution of proposed longitudinal-torsional transducer model to the application in mechanical engineering or other disciplines? In is expected that the authors briefly emphasize this point in the Conclusions section.

Response: Thank you very much for your valuable comments. We have partially modified the conclusions with a relevant formulation and summary of the contribution of the proposed longitudinal torsional transducer to the discipline, and you can find the corresponding items in Section 5.

On page 20:This paper proposed a longitudinal-torsional transducer for honeycomb aramid material during ultrasonic vibration assisted milling (UVAM), and the directivity of transducer was analyzed by simulation and experiment, which provides a new idea for the processing of honeycomb aramid material and its widely application. The major conclusions were listed as the following:

Round 2

Reviewer 1 Report

Revision is in the good way but I do not see the booster ideas given by Celaya in Ultrasonic Assisted Turning of mild steels,International journal of materials & product technology 37 (1), 60

Reviewer 2 Report

Required corrections have been made.